# Powder-in-Tube Reactive Molten-Core Fabrication of Glass-Clad BaO-TiO_2_-SiO_2_ Glass–Ceramic Fibers

**DOI:** 10.3390/ma13020395

**Published:** 2020-01-15

**Authors:** Shuo Yang, Hanna Heyl, Daniel Homa, Gary Pickrell, Anbo Wang

**Affiliations:** 1Center for Photonics Technology, The Bradley Department of Electrical and Computer Engineering, Virginia Polytechnic Institute and State University, Blacksburg, VA 24060, USA; 2Department of Material Science and Engineering, Virginia Polytechnic Institute and State University, Blacksburg, VA 24060, USA

**Keywords:** thermal drawing, molten core fabrication, glass–ceramic fiber, second-order harmonic generation, photoluminescence

## Abstract

In this paper we report the fabrication of glass-clad BaO-TiO_2_-SiO_2_ (BTS) glass–ceramic fibers by powder-in-tube reactive molten-core drawing and successive isothermal heat treatment. Upon drawing, the inserted raw powder materials in the fused silica tubing melt and react with the fused silica tubing (housing tubing) via dissolution and diffusion interactions. During the drawing process, the fused silica tubing not only serves as a reactive crucible, but also as a fiber cladding layer. The formation of the BTS glass–ceramic structure in the core was verified by micro-Raman spectroscopy after the successive isothermal heat treatment. Second-harmonic generation and blue-white photoluminescence were observed in the fiber using 1064 nm and 266 nm picosecond laser irradiation, respectively. Therefore, the BTS glass–ceramic fiber is a promising candidate for all fiber based second-order nonlinear and photoluminescence applications. Moreover, the powder-in-tube reactive molten core method offers a more efficient and intrinsic contamination-free approach to fabricate glass–ceramic fibers.

## 1. Introduction

Since the introduction of optical fibers in the 1960s, optical fibers have shown an increased demand in various applications, such as in communications, lasers, and sensing, owing to its compact size, strong light confinement, and long interaction length [1,2,3]. However, due to the amorphous nature of glass, optical glass fibers generally do not exhibit second-order optical nonlinearity, which is crucial to many devices, such as frequency doublers, electro-optic modulators, and the generation of entangled photon pairs [4]. Efforts have been made to overcome this obstacle by using thermal poling of glass fibers [5,6,7] and the fabrication of crystalline fibers [8,9,10]. The first approach induces an effective second-order nonlinearity by the combination of a frozen-in DC field and intrinsic third-order nonlinearity. Since it relies on higher-order nonlinearity and non-uniform distribution of ions, the induced second-order nonlinearity is usually relatively weak and unstable upon heat or strong light irradiation. The second approach fabricates common nonlinear crystals into the fiber, such as LiNbO_3_, BBO (beta barium borate) or KDP (potassium dihydrogen phosphate), and the resulting fibers show strong and stable second-order nonlinearity as their bulk forms. However, since the process generally involves crystal growth from the melt, the yield tends to be low [8,9,10]. Usually, the fiber either does not have a cladding layer or exhibits strong elemental migration if a cladding layer exists [8]. In the past decades, glass–ceramic materials have emerged as a potential candidate to fill the gap, since it is a two-phase composite in which crystals are homogeneously distributed in the glass matrix [11,12]. Among them, glass–ceramic materials with fresnoite crystals (i.e., BaO-TiO_2_-SiO_2_ (BTS)) have been extensively studied owing to their good dielectric and optical properties [13,14,15,16,17]. Besides strong second-order nonlinearity [13,14,15], long-decaying UV photoluminescence [16,17] has also been demonstrated in BTS crystals. Hence, BTS glass–ceramic fibers are a promising candidate for second-order nonlinear and photoluminescence applications in an all-fiber-based system. To date, only a few fresnoite glass–ceramic fibers have been reported due to their difficult fabrication [18,19,20]. Direct drawing of glass–ceramic fibers is challenging due to the crystallization of the preform at the drawing temperature [20]. Hane et al. used the pull-from-melt method, in which the fabricated fiber does not have a cladding and exhibits large diameter variations [18]. Ohara et al. and Fang et al. overcame this obstacle by using “rod-in-tube” molten-core thermal drawing [19,20]. However, in both processes the precursor core glass needs to be synthesized first via the “melt-and-quench” method and then machined to the desired geometry. In the glass melting process, a platinum (Pt) crucible is often used to prevent reactions between the glass melt and the crucible. However, since the temperature required to synthesize BTS glass is high (~1600 °C), the undesired dissolution of the crucible material is promoted. For example, dissolution of platinum (Pt) from the crucible into the glass melt and the reduction of transparency and change of crystallization behavior have been reported [21]. In addition, exposure to the machining tools also raises the risk of contamination.

On the other hand, reactive molten-core fabrication has proven to be an effective method to fabricate fibers with core materials which cannot be drawn into fibers on their own [22,23,24]. With the housing material reacting with the core material via various mechanisms, such as reactive chemistry, dissolution, and diffusion, fibers with various core materials, such as semiconductors, special glass, and crystalline oxides, have been reported [22,23,24,25,26,27,28,29,30]. Therefore, in this paper we report on the fabrication of glass-clad BTS glass–ceramic fibers using the powder-in-tube reactive molten-core approach with a successive isothermal heat treatment. The corresponding crystalline powder mixture is filled inside a glass tube as the fibers preform, and the precursor core glass is directly synthesized due to reactions occurring during the fiber drawing process. The housing glass not only serves as a material source for the synthesis of the core glass, it also serves as the cladding layer of the drawn fiber. This method utilizes the reaction between the melt and the crucible, which avoids the potential occurrence of undesired reactions during the synthesis of the precursor glass by the “melt-and-quench” approach in previous works [19,20]. It is also more efficient and can be extended to fabricate other glass–ceramic fibers of which compositions contain the housing material. Second-harmonic generation and photoluminescence properties were also measured in the fabricated fiber.

## 2. Materials and Methods 

### 2.1. Preform Assembly and Fiber Fabrication

Figure 1 shows the preform assembly and fabrication procedure. A glass tubing (ID: 3 mm, OD: 8 mm) was sealed on one end and then over-cladded consecutively by two other tubes (ID: 9 mm OD: 15 mm and ID: 16 mm, OD: 20 mm) with a glass working lathe (Litton Engineering Laboratories, Grass Valley, CA, USA). The glass tubes used were fused quartz (Technical Glass Product) and the final ID and OD of the preform was 3 mm and 19 mm, respectively. The raw materials used were BaCO_3_ powder 99.8% (Alfa Aesar, Tewksbury, MA, USA) and TiO_2_ powder 99.5% (Sigma-Aldrich, St. Louis, MO, USA). The powders were prepared in a mole ratio of 70 BaCO_3_: 30 TiO_2_ and thoroughly mixed in acetone inside an ultrasonic bath. This composition was chosen because of its effectiveness to form a BTS glass–ceramic structure, as reported in the literature [13,31,32,33,34]. The mixed power was inserted into the preform and the preform was then heated up to ~1900 °C in a tube furnace under a nitrogen environment, while the inside of the tube was kept in a low vacuum (~0.1 bar). During the heating process, BaCO_3_ starts to decompose into BaO and CO_2_ at ~900 °C, and becomes fully decomposed at ~1400 °C [35]. The powder was not compressed firmly in the tube, so as to ensure no build-up of pressure during the release of CO_2_ gas in the powder state. During this process, the resulting BaO reacts with TiO_2_ (powder) and SiO_2_ (tubing) to form the BTS compound. The BaO and TiO_2_ themselves do not melt until 1923 °C and 1843 °C, respectively. However, the BTS compound has an eutectic point of ~1260 °C and a melting point of ~1445 °C [13]. Therefore, the molten BTS compound dissolves the remaining BaO, TiO_2_ and SiO_2_ during the process. In addition, owing to the low vacuum environment, CO_2_ was easily released from the melt due to the buoyancy effect. To prevent the early collapse of the preform and a change in the drawing condition once the temperature passed 1600 °C, the vacuum pump was shut off and the inside of the tube was exposed to air at 1 atmosphere pressure. At the drawing temperature (~1900 °C), due to the small size and the liquid state, the core composition is homogenized via fast diffusion and convection induced by the thermal gradient [36]. Once the fused silica is soft enough, the preform is drawn into a fiber at speeds of 15–20 cm/s. The drawn fiber has a diameter of 300 μm to 400 μm depending on different drawing speeds. The drawn fiber was then isothermally heat treated at 850 °C for 10 h with a heating rate of 10 °C/min to form a glass–ceramic core. This specific thermal heat treatment was chosen based on the composition analysis of the fiber (which will be discussed in the Section 3) and its corresponding crystallization temperature, reported in the literature [13,31,32,33,34].

### 2.2. Characterization

The fiber geometry was examined using a transmission optical microscope (Olympus BX51, 20× Objectives, Center Valley, PA, USA) and an environmental scanning electron microscope (ESEM, FEI Quanta 600 FEG, Hillsboro, OR, USA). The elemental mapping and line scan were performed on a polished fiber cross-section (down to a 0.1 μm diamond polishing pad) at an accelerating voltage of 20 kV using the same ESEM with an attached energy dispersive X-ray spectrometer (Bruker QUANTAX 400, Billerica, MA, USA) equipped with a high-speed silicon drift detector. 

Micro-Raman spectroscopy analyses were performed on a polished fiber cross-section with an unpolarized confocal Raman microscope (WITec alpha-300-SR, Ulm, Germany). A continuous wavelength (CW) laser of 532 nm was focused onto the sample through a 50× objective and the spot size was ~2.5 µm. The reflected light was filtered by a long-pass filter (cut-off: 535 nm) before entering the spectrometer equipped with an electron-multiplying CCD spectroscopy detector (Andor Newton DU970, Abingdon, UK). The exposure time was 1 s.

The second-order nonlinearity of the fabricated fiber was investigated by irradiating the fiber core with a pulsed laser operating near 1064 nm with a pulse width of ~8 ps (Passat COMPILER, Vaughan, ON, Canada). In the measurement, the pulse energy varied but the repetition rate was fixed at 400 Hz. The laser beam was focused onto the fiber core with a 10× objective lens and the transmitted light from the other end of the fiber was recorded by a spectrometer (Ocean Optics, USB4000-VIS-NIR, Largo, FL, USA) after the pump wavelength was filtered. The spectra were recorded at 0.5 s exposure time and averaged from 100 recorded data. The photoluminescence (PL) property of the fabricated fiber was also investigated by similar configurations with the same laser operating at 266 nm (4th harmonics) with an averaging power of 800 μW. The laser beam was carefully tuned to irradiate just a portion of the fiber core to minimize the interference from the emission of the cladding. The spectra of the transmitted light on the other end of the fiber were recorded by the same spectrometer.

## 3. Results

Representative optical microscopic and SEM images of the fabricated fiber are shown in Figure 2. The fabricated fiber has a diameter of ~300 μm with a ~25 μm brown-transparent BTS glass–ceramic core (Figure 2a). The core color is due to the absorption of the Ti^4+^-O^2−^ charge transfer in the system [37]. The refractive index of the BTS glass–ceramics ranges from 1.6 to 1.9 under different compositions and crystallization levels, which is higher than silica (~1.45) [13]. Therefore, the structure of this fiber forms a waveguide. The core-cladding ratio of the fiber is about 1:12, which is smaller than the initial preform (~1:6.3). This is due to the significant difference of viscosity between the core and the cladding material, which shows a different fluid dynamic behavior during the drawing. Overall, both the core and cladding exhibit good circularity, as shown in Figure 2b. Some micro-deformations were observed at the core/cladding boundary, as indicated by Figure 2c,d. This is most likely due to the incomplete filling of the glass tube after the powders have melted, which induces the non-uniform collapse of the housing glass during the thermal drawing process. This irregularity can be reduced by optimizing the preform dimension, powder filling process, powder size, and drawing parameters. No crack at the core-cladding interface, as well as trapped gas in the core, was observed in the fiber. Figure 2c,d present the elementary mapping of Si, Ba, and Ti of the core measured by EDX. Since Ba and Ti have a spectral overlapping in the EDX spectrum due to the limited energy resolution, it is hard to derive their ratio accurately. Instead, the ratio of Ba and Ti to Si can be estimated by characterizing the intensity of the Ba_L_ + Ti_K_ and Si_K_ channels in the EDX spectrum [38]. Since there is no report of strong vaporization of Ba or Ti in the BTS system under the drawing condition [39], it is safe to assume that the concentration of Ba and Ti in the fiber preserves its ratio in the initial powder mixture. As shown in Figure 2c,d, Ba and Ti are concentrated in the core where Si has a steep decrease. The boundary in EDX results matches the core observed in Figure 2b using Back-Scattered-Electron (BSE) imaging.

To offer a better insight on the composition, a line scan was performed for fibers with different core diameters, which were tuned by changing the drawing temperature and speed. Representative results of the core sizes of 25 μm and 50 μm are plotted in Figure 3. The results indicate that both Si, Ba, and Ti have a uniform distribution inside the core and a significant diffusion-based profile at the core/cladding interface. Since the Si presented in the core comes purely from the reaction between the core material and the fused silica tubing during the drawing process, the average Si content inside the core with different core sizes are summarized in Table 1. The results indicate that larger core diameters have less Si inside the core, which can be explained by the longer diffusion length of the Si in the core [24,36]. For all the fabricated fibers with different core sizes, the molar ratio of Si is within 65–75 mol%. Owing to the large glass formation range for BTS in terms of the SiO_2_ composition (~40–80 mol% [40]), the fiber core can still form glass upon drawing without crystallization. 

For the characterization of the fiber’s microstructure, a micro-Raman spectroscopy of the fibers’ cross-section was measured, and the results are shown in Figure 4. The Raman spectrum of the as-drawn fiber core consists of several broad bands. The broad band located in the 200 to 500 cm^−1^ range corresponds to the Si-O-Si bending vibration modes of the SiO_4_ unit [41]. The band ranging from 800 to 1000 cm^−1^ can be associated with the stretching mode of the Ti-O bond [41]. The absence of sharp peak in the spectrum confirms that the as-drawn fiber core is amorphous. For the Raman spectrum of the isothermally heat-treated fiber core, several sharp peaks are observed. A strong peak at ~865 cm^−1^ is observed and is assigned to the vibration of the short Ti-O bond [20,41,42,43]. Three peaks, at 531, 598, and 663 cm^−1^, are also present in the spectrum, which are assigned to the ν(TiO_4_) and ν_s_(Si-O-Si) modes [20,42]. Moreover, there are several weak peaks at 221, 273, and 343 cm^−1^, which can be attributed to the translational and bending modes of the Si_2_O_7_ and TiO_5_ groups [20,42]. All the observed sharp peaks are in good agreement with the previously reported Raman spectrum of a Ba_2_TiSi_2_O_8_ single crystal [20,41,42,43], confirming crystallization after the thermal heat treatment. The Raman spectrum of the cladding region after the isothermal heat treatment is also presented in Figure 4, which shows a characteristic Raman spectrum of silica glass [20]. The results indicate that crystallization is confined inside the fiber core.

The results of the second-order harmonic generation are shown in Figure 5. The as-drawn fiber without thermal heat treatment does not show any detectable emission within the measurement range of the spectrometer. In contrast, the isothermally heat-treated fiber exhibits a strong green emission with narrow bandwidth centered at 531.4 nm, which is exactly half of the pump wavelength. An image of the green emission from the fiber, taken by a DSLR camera, is shown as the insert in Figure 5a. According to [20], the energy from an infrared pulse is not sufficient to induce such an emission under a single-photon process, which indicates that this narrow visible emission is at least a two-photon process, most likely the second-harmonic generation. The relationship between the emission intensity and the irradiation power was measured, and the result plotted in double-logarithmic scale in the insert in Figure 5a. In a nonlinear process, the emission intensity is proportional to the *n*-th power of the irradiation power where *n* is the order of the nonlinear process [4]. Thus, a linear fitting line of the experimental data is also plotted, and the slope is 2.28, which confirms that the emission is attributed to the second-harmonic generation [20].

It has been reported that BTS glass–ceramics exhibit a broad blue-white luminescence in the range of about 400 to 600 nm under UV light exposure [16,44]. Thus, it is expected that the fiber fabricated in this study will also show similar luminescent properties. The recorded photoluminescence (PL) spectra excited at 266 nm of the as-drawn and the thermally heat-treated fiber are shown in Figure 5b. Both spectra have a range of 400 to 700 nm, in which two peaks can be observed. The as-drawn fiber core exhibits a broad band emission in the visible range with a main peak at ~530 nm, and the thermally heat-treated fiber shows a similar emission spectrum with a stronger intensity and a blue-shifted main peak position of ~500 nm. The enhancement of PL intensity and the shift of the peak position after the heat treatment match the results reported in [44,45]. The origin of the PL in the BTS glass–ceramics is controversial [44], but most of the work reported in the literature [16,44,45,46,47] attributes it to the TiO_5_ pyramid within the BTS crystals. The origin of the weak peak of ~450 nm needs further investigation, but it might be attributed to the Ti^4+^ impurities or/and oxygen-related defects of the SiO_4_ units in the glass network, as suggested by [44,46]. Note that, due to the overlapping of the emission spectra from different color centers, the observed peak positions in this study can be slightly shifted to the actual positions and, hence, can differ from the values reported in other references [44,45,46,47]. Different processing methods may also affect the PL spectrum, as suggested by [44]. Thus, further investigation is ongoing to gain a better understanding of the PL property in the fabricated fibers. An image of the blue-white emission from the thermally heat-treated fiber, taken by a DSLR camera, is also shown as the insert in Figure 5b.

## 4. Conclusions

In conclusion, we demonstrated the feasibility of using the reactive molten-core method to fabricate BTS glass–ceramic fibers. During this method, raw powder materials were inserted inside a customized fused quartz tubing in which the powder reacted with the housing tubing to form a BTS compound upon fiber drawing. The fiber was drawn at high speeds to prevent any uncontrollable crystallization during the cooling followed by isothermal heat treatment at 850 °C for 10 h to form a glass–ceramic core. Micro-Raman spectroscopy confirmed the transition from an amorphous glass into a glass–ceramic core after the heat treatment, with highly localized crystallization inside the core. The fabricated fiber exhibited second-order nonlinearity through the second-harmonic generation using a picosecond pulse laser. The fiber also showed broad photoluminescence covering all the visible spectrum upon irradiation with the 266 nm laser, and its spectrum was enhanced and blue-shifted after the thermal heat treatment. Therefore, BTS glass–ceramic fibers provide a promising opportunity for all-fiber-based second-order nonlinear and photoluminescence applications. Moreover, the reactive molten-core method offers a more efficient and contamination-free approach to fabricate glass–ceramic fibers.

## Figures and Tables

**Figure 1 materials-13-00395-f001:**
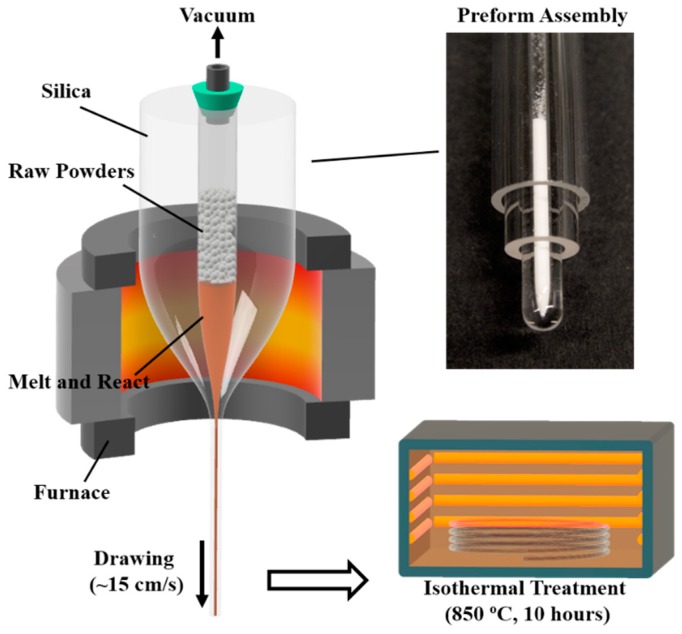
Preform assembly and fabrication procedure of BTS glass–ceramic fibers (BTS: BaO-TiO_2_-SiO_2_).

**Figure 2 materials-13-00395-f002:**
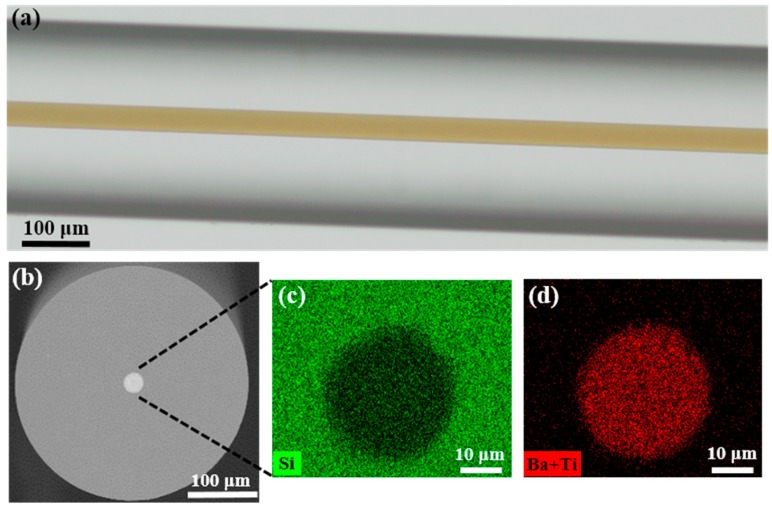
(**a**) The optical transmitted microscopic image of the fabricated BTS glass–ceramic fiber which has a transparent-brown core; (**b**) BSE image of the cross section of the fiber; (**c**,**e**) EDX mapping of elements Si, Ba, and Ti in the core. (BSE: Back scattered electrons, EDX: Energy-dispersive X-ray spectroscopy).

**Figure 3 materials-13-00395-f003:**
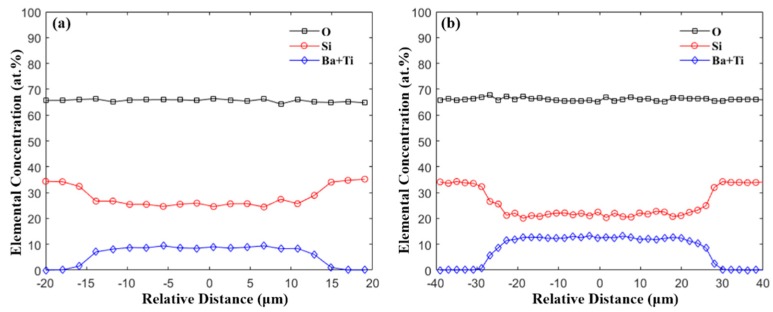
EDX line-scan across the fiber core with core diameter of (**a**) 24.6 μm and (**b**) 50.8 μm.

**Figure 4 materials-13-00395-f004:**
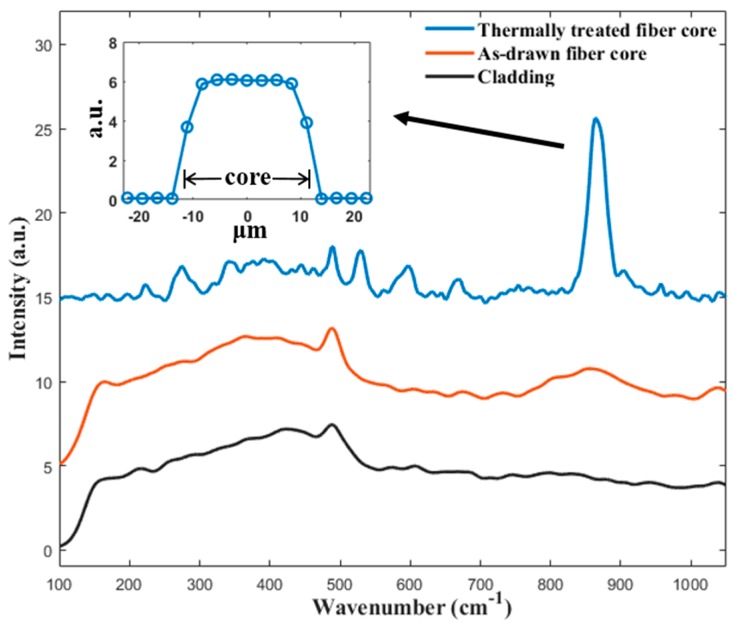
Raman spectra of the fabricated BTS fiber before and after the thermal heat treatment. The insert shows the line scan profile of the peak strength at 865 cm^−1^ across a thermally heat-treated fiber core.

**Figure 5 materials-13-00395-f005:**
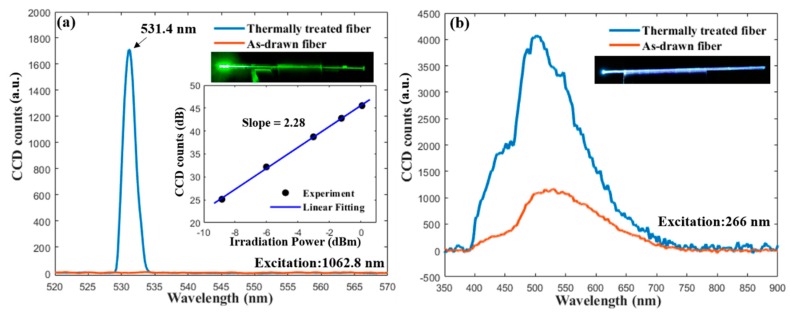
(**a**) The emission spectra of the as-drawn and thermally heat-treated fiber excited by a 1064 nm picosecond laser. The inserts show an image of the emission taken by a DSLR camera and the relation between the emission intensity and irradiation power in double-logarithmic scale; (**b**) the photoluminescence spectra of the as-drawn and thermally heat-treated fiber excited by a 266 nm laser.

**Table 1 materials-13-00395-t001:** Average elemental concentration inside core with different core sizes.

Core Diameter (μm)	Ba + Ti (at.%)	Si (at.%)
24.6	8.7 ± 0.4	25.6 ± 0.8
50.8	12.3 ± 0.6	21.6 ± 0.8
113.8	11.8 ± 0.6	21.8 ± 0.7

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
