# Peer review of "Powder-in-Tube Reactive Molten-Core Fabrication of Glass-Clad BaO-TiO2-SiO2 Glass–Ceramic Fibers"

_materials, 2020, doi:10.3390/ma13020395_

Round 1

Reviewer 1 Report

The manuscript is extremely well written and the results are very well presented. The paper demonstrates the feasibility of using the reactive molten-core method to fabricate BaO-TiO2-SiO2 glass-ceramic fibers. The methodology applied prevents uncontrolled crystallization of the sample during the drawing/cooling process, allowing the post-synthetic crystallization of the desired phase by isothermal heat-treatment. I fully recommend the publication of this work in Materials. I have only two minor questions:

The authors cite a work (Ref. 20 in the manuscript), which reports glass-ceramic optical fiber containing Ba2TiSi2O8 nanocrystals with similar properties. Is there a way of comparing the efficiency of both systems? For example taking in consideration the frequency conversion efficiency (named h in Ref. 20). Lines 176-183: It is expected that the emission at centered at 531.4 nm under excitation at 1062.8 nm is generated by a two-photon process. Nevertheless, the authors should further prove this fact by analyzing the relationship between the emission intensity and the irradiation power, such as performed in Ref. 20 (page 6).

Author Response

Thank you so much for your review of our paper. We have answered each of your points below.

(1) The authors cite a work (Ref. 20 in the manuscript), which reports glass-ceramic optical fiber containing Ba2TiSi2O8 nanocrystals with similar properties. Is there a way of comparing the efficiency of both systems? For example taking in consideration the frequency conversion efficiency (named h in Ref. 20). Lines 176-183:

Response: We thank the reviewer for this comment. In order to characterize the efficiency, we need to measure the output power of the second-harmonic generation, which is also susceptive to the fiber propagation loss. This work focuses on demonstrating the feasibility to use the powder-in-tube reactive molten-core method to fabricate glass-ceramic fiber. We will investigate the fiber properties, including absorption, propagation loss, and the efficiency of the second-harmonic generation, which will be presented in our future work.

(2) It is expected that the emission at centered at 531.4 nm under excitation at 1062.8 nm is generated by a two-photon process. Nevertheless, the authors should further prove this fact by analyzing the relationship between the emission intensity and the irradiation power, such as performed in Ref. 20 (page 6). 

Response: We thank the reviewer for this comment. The relation between the emission intensity and the irradiation power were measured and the result is plotted in double-logarithmic scale in the insert in Figure 5(a). In a nonlinear process, the emission intensity is proportional to the n-th power of the irradiation power where n is the order of the nonlinear process [4]. Thus, a linear fitting line of the experimental data is also plotted and the slope is 2.28, which confirms that the emission is attributed to the second-harmonic generation [20]. The corresponding description has been added in lines 190-195.

Reviewer 2 Report

The manuscript describes a new technique to make a glass ceramics fiber. The novel fiber made shows second order nonlinearity. Both the new fabrication technique and the demonstration of a fiber with second order nonlinearity make the manuscript highly significant and thus well suited for publication. The manuscript is well written and the figures are well presented. The authors should address the following point before the paper can be accepted for publication.

* In the Introduction, the authors distinguish between melt-in-tube and molten core fabrication, but both are essentially the same as in both cases the core material is in the molten stage within a housing tube when drawn. The authors should clarify this or only use one term for the same thing.

* Further to the point above, it seems the distinguishing feature between what the authors call melt-in-tube and molten core technique is that the melt-in-tube example given in the manuscript uses a rod as starting material, whereas the molten core technique used by the authors uses crystalline raw materials as starting material. This should be made clear, in particular in the last paragraph of the Introduction, it should be mentioned that crystalline raw materials are used as the core starting material.

* The authors argue that their method avoids impurities from the crucible material by giving Pt impurities from Pt crucible corrosion as example. This is not a reasonable argumentation as in the case of the manuscript, the crucible (i.e. housing tubing) is severely corroded by the glass batch during melting. The difference is that in the case of the Pt crucible, the corrosion/dissolution is undesired, whereas in the case of the silica tubing the dissolution is desired. One could get around the Pt contamination problem by using a silica or alumina crucible. However, this is often not done as the dissolution of silica or alumina from the crucible is undesired, which is different to the case in this manuscript.

* In the paragraph below figure 3, the core color is attributed to Ti4+ units [37]. According to Ref [37], the color is due to Ti4+ - O2- charge transfer, i.e. not from Ti4+ units. The authors should correct this.

* The discussion of photoluminescence (PL) is confusing; first it is attributed to oxygen defects, then it is attributed to TiO5 pyramid. According to Ref [44], the origin of the PL is controversial. If the authors want to convey this and thus give different causes for the PL, then this needs to be clearly articulated. Furthermore, according to Ref[44], the oxygen defect is in the SiO4 unit and not in the Ti-O polyhedral. The authors should carefully read Refs [44,45] to make sure these references are correctly cited.

* The authors attribute the weak PL at 450nm to Ti4+. The authors should check if this is correct or if the weak PL is due to Ti3+.  

Author Response

Thank you so much for your review of our paper. We have answered each of your points below.

(1) In the Introduction, the authors distinguish between melt-in-tube and molten core fabrication, but both are essentially the same as in both cases the core material is in the molten stage within a housing tube when drawn. The authors should clarify this or only use one term for the same thing.

Response: Thank you very much for the comment. We rephrased the work done by others [19, 20] as “rod-in-tube” and ours as “powder-in-tube” to clarify the difference. The manuscript has been revised accordingly in lines 52-53.

(2) Further to the point above, it seems the distinguishing feature between what the authors call melt-in-tube and molten core technique is that the melt-in-tube example given in the manuscript uses a rod as starting material, whereas the molten core technique used by the authors uses crystalline raw materials as starting material. This should be made clear, in particular in the last paragraph of the Introduction, it should be mentioned that crystalline raw materials are used as the core starting material.

Response: Thank you very much for the suggestion. Followed by the previous point, we also emphasized that the powder used in our approach are crystalline materials rather than glass. The manuscript has been revised accordingly in lines 65-69.

(3) The authors argue that their method avoids impurities from the crucible material by giving Pt impurities from Pt crucible corrosion as example. This is not a reasonable argumentation as in the case of the manuscript, the crucible (i.e. housing tubing) is severely corroded by the glass batch during melting. The difference is that in the case of the Pt crucible, the corrosion/dissolution is undesired, whereas in the case of the silica tubing the dissolution is desired. One could get around the Pt contamination problem by using a silica or alumina crucible. However, this is often not done as the dissolution of silica or alumina from the crucible is undesired, which is different to the case in this manuscript.

Response: Thank you very much for the comment. The argument about the avoidance of impurities was based on the following fact. In previous reported works [19, 20], the core glass rod is synthesized first via “melt-in-quench” method and a Pt crucible is often used in this case to melt the glass to prevent the reaction with the crucible. However, since the temperature required to synthesize BTS glass is high (~1600°C), the undesired dissolution of crucible material is promoted and reported in [21]. On the other hand, our method uses the desired reaction to synthesize the core glass during the fiber drawing process, and thus, avoids the undesired reaction which may occur in the method used in previous reported works.

However, the reviewer’s point is well taken and we have changed our argument in the following expression for clarification: Our method utilizes the reaction between the melt and crucible, which avoids the potential occurrence of undesired reaction during the synthesis of precursor glass by “melt-and-quench” approach in previous works. The manuscript has been revised accordingly in lines 54-58, 71-74.

(4) In the paragraph below figure 3, the core color is attributed to Ti4+ units [37]. According to Ref [37], the color is due to Ti4+ - O2- charge transfer, i.e. not from Ti4+ units. The authors should correct this.

Response: Thank you very much for the correction. The manuscript has been revised accordingly in line 133.

(5) The discussion of photoluminescence (PL) is confusing; first it is attributed to oxygen defects, then it is attributed to TiO5 pyramid. According to Ref [44], the origin of the PL is controversial. If the authors want to convey this and thus give different causes for the PL, then this needs to be clearly articulated. Furthermore, according to Ref[44], the oxygen defect is in the SiO4 unit and not in the Ti-O polyhedral. The authors should carefully read Refs [44,45] to make sure these references are correctly cited.

Response: Thank you very much for the comments and suggestions. We have carefully reviewed those two and additional references. We rephrased the discussion of photoluminescence (PL) for better clarity as shown below:

The as-drawn fiber core exhibits a broad band emission in the visible range with a main peak at ~530 nm and the thermally heat-treated fiber shows a similar emission spectrum with a stronger intensity and blue-shifted main peak position ~500 nm. The enhancement of the PL intensity and the shift of the peak position after the heat-treatment match the results reported in [44, 47]. The origin of the PL in BTS glass-ceramics is controversial [44] but most of the work reported in literature [16, 44-47] attributes it to the TiO5 pyramid within BTS crystals. The origin of the weak peak ~450 nm needs further investigation but it might be attributed to the Ti4+ impurities or/and oxygen-related defects of SiO4 units in the glass network as suggested by [44, 45].

The manuscript has been also revised accordingly in lines 200-213.

(6) The authors attribute the weak PL at 450nm to Ti4+. The authors should check if this is correct or if the weak PL is due to Ti3+.  

Response: Thank you very much for the comments. As suggested in [44], the origin of the PL in the BTS glass-ceramic is controversial. Thus, further investigation is ongoing to gain better understanding of the PL property in the fabricated fiber. In the current stage, we are inclined to attribute this weak peak to the Ti4+ unit or/and oxygen-defects of SiO4 units in the glasses given the matched positions to the results reported in the similar material system [44, 45]. To the best of our knowledge, there is no reported PL property that originated from Ti3+ in the BTS system. In addition, the PL property originating from Ti3+ ions in other reported silicate glass systems exhibits the emission spectrum in the range of 500-700 nm which is a much longer range than the one observed in our experiment (refer to the following references). Hence, Ti3+ ions are unlikely to be the origin of the observed weak PL.

References:

[1] Dianov, E.M.; Lebedev, V.F.; Zavorotnyi, Y.S., Luminescence of Ti3+ ions in silica glass. Quantum Electronics 2001, 31, 187.

[2] Song, C. F.; M. K. Lü; P. Yang; D. Xu; D. R. Yuan., Study on the photoluminescence properties of sol-gel Ti3+ doped silica glasses, Journal of sol-gel science and technology 2002, 25, 113-119.

Reviewer 3 Report

In this paper the authors presented a technique to produce BTS glass-ceramic fibers that exhibit second-harmonic generation and white-blue photoluminescence. The results are very interesting and are worth to be published however, there are some issues that require some attention. First, Fig. 2a show some irregularities (microdeformations) in the fiber diameter (core and cladding) that are not mentioned by the authors. Second, is it possible to produce, by using this technique, a BTS glass-ceramic fiber with dimensions compatible with standard fibers (9/125 mm)? and which were the oven temperatures to produce the fibers with different core diameters? Finally, there is no characterization of the spectral absorption/attenuation of the produced fiber. A more careful reading is also required in order to overcome some typo errors.

Author Response

Thank you so much for your review of our paper. We have answered each of your points below.

(1) First, Fig. 2a show some irregularities (microdeformations) in the fiber diameter (core and cladding) that are not mentioned by the authors.

Response: We thank the reviewer for this comment. It is most likely due to the incomplete filling of the glass tube after the powders melt, which induces the non-uniform collapse of the housing glass during the thermal drawing process. This irregularity can be reduced by optimizing the preform dimension, powder filling process, powder size, and drawing parameters. Those descriptions have been added to the manuscript in lines 139-143.

(2) Second, is it possible to produce, by using this technique, a BTS glass-ceramic fiber with dimensions compatible with standard fibers (9/125 mm)? and which were the oven temperatures to produce the fibers with different core diameters?

Response: We thank the reviewer for this comment. Yes, it is possible. Note that, as indicated in Table 1, a smaller core diameter increases the Si content, which will alter the crystallization behavior and fiber property. We are currently investigating the effect of the fiber core diameter and the drawing parameters on the fiber properties.

For the second part of the question, we usually keep the furnace temperature relatively stable and tune the preform dimension, the preform feeding rate, and the fiber drawing rate to control the final fiber dimension.

(3) Finally, there is no characterization of the spectral absorption/attenuation of the produced fiber.

Response: Thank you very much for the comment. This work focuses on demonstrating the feasibility to use the powder-in-tube reactive molten-core method to fabricate glass-ceramic fibers. We are currently investigating the fiber properties, including the absorption spectrum, the attenuation, and the micro-structure, and also the influence of drawing parameters (i.e. drawing rate, fiber core diameter), the initial powder composition, and the thermal treatment condition on those fiber properties. The results will be presented in future publications.

(4) A more careful reading is also required in order to overcome some typo errors.

Response: We thank the reviewer for this comment. We have carefully reviewed the manuscript and corrected all the typos we could find.
